# A generalized epilepsy network derived from brain abnormalities and deep brain stimulation

Gong-Jun Ji [1,2,3,4,5,33], Michael D. Fox [6,33], Mae Morton-Dutton[6], Yingru Wang[4], Jinmei Sun [1,3], Panpan Hu[1,3], Xingui Chen[1,3], Yubao Jiang[1,3], Chunyan Zhu [4], Yanghua Tian [2,3,4], Zhiqiang Zhang [7], Haya Akkad[6,8], Janne Nordberg [9,10], Juho Joutsa [9,10], Cristina V. Torres Diaz[11], Sergiu Groppa [12], Gabriel Gonzalez-Escamilla [12], Maria de Toledo[13], Linda J. Dalic[14], John S. Archer [15], Richard Selway[16], Ioannis Stavropoulos [17,18], Antonio Valentin[17,18,19], Jimmy Yang[20,21], Faical Isbaine[22], Robert E. Gross[23], Sihyeong Park [24], Nicholas M. Gregg [24], Arthur Cukiert [25], Erik H. Middlebrooks[26], Nico U. F. Dosenbach [27,28,29], Joseph Turner [6], Aaron E. L. Warren [6], Melissa M. J. Chua[6], Alexander L. Cohen [6,15], Sara Larivière[6], Clemens Neudorfer [6,30], Andreas Horn [6,30], Rani A. Sarkis [6], Ellen J. Bubrick[6], Robert S. Fisher[31], John D. Rolston [6], Kai Wang [1,3,4,5,32,33] ✉ & Frederic L. W. V. J. Schaper [6,33] ✉

Idiopathic generalized epilepsy (IGE) is a brain network disease, but the location of this network and its relevance for treatment remain unclear. We combine the locations of brain abnormalities in IGE (131 coordinates from 21 studies) with the human connectome to identify an IGE network. We validate this network by showing alignment with structural brain abnormalities previously identified in IGE and brain areas activated by generalized epileptiform discharges in simultaneous electroencephalogram-functional magnetic resonance imaging. The topography of the IGE network aligns with brain networks involved in motor control and loss of consciousness consistent with generalized seizure semiology. To investigate therapeutic relevance, we analyze data from 21 patients with IGE treated with deep brain stimulation (DBS) for generalized seizures. Seizure frequency reduced a median 90% after DBS and stimulation sites intersect an IGE network peak in the centromedian nucleus of the thalamus. Together, this study helps unify prior findings in IGE and identify a brain network target that can be tested in clinical trials of brain stimulation to control generalized seizures.

Up to one-third of all patients with epilepsy have idiopathic generalized epilepsy (IGE)[1]. Patients with IGE have one or more generalized-onset seizure types, including generalized tonic-clonic, absence, and/or myoclonic seizures, among others[2,3]. Up to 27% of patients experience seizures despite antiseizure drug treatment highlighting the need for new therapies[4]. IGE is increasingly conceptualized as a brain network disease[5–11], however the location of this network and its therapeutic relevance remain unclear[12,13].

Patients with IGE often show generalized spikes or spike-wave (GSW) discharges on electroencephalogram (EEG), while having "normal" magnetic resonance imaging (MRI). However, subtle brain abnormalities can be detected at the group level using structural and functional neuroimaging, including gray matter atrophy[6,14–20] and increased spontaneous local activity measured with resting-state functional MRI (rs-fMRI)[21–23]. Different studies have identified abnormalities across different brain regions, leaving the neuroanatomical basis of IGE unclear[19,24–26]. Identifying the regions and networks these abnormalities map to may help pinpoint an IGE network and inform a potential target for therapeutic intervention.

We recently developed a technique termed 'coordinate network mapping,' which tests whether heterogeneously distributed neuroimaging abnormalities map to a common brain network[27]. Coordinate network mapping utilizes an atlas of human brain connectivity (i.e., a human connectome) to identify the network of brain regions functionally connected to a given set of coordinates. This technique is an adaptation of lesion network mapping[28–31] and replaces the lesion locations with coordinates of neuroimaging abnormalities. Coordinate network mapping has helped identify common brain networks that link heterogeneous neuroimaging findings in neurodegenerative diseases, migraine, addiction, and several other neuropsychiatric conditions[27,32–37]. These coordinate networks can then be combined with information from brain stimulation to investigate the therapeutic relevance of the findings[38], providing multimodal support for network localization[30,39,40].

In this study, we combined coordinates of brain abnormalities and deep brain stimulation (DBS) data with the human connectome to identify a generalized epilepsy network.

## Results

### Systematic search and coordinates

We identified a total of 767 publications using a systematic search, of which 81 full texts were assessed based on relevance (See Online Methods, Supplementary Fig. 1). We included 20 publications describing 21 independent studies (540 patients with IGE and 778 healthy controls) identifying coordinates of structural and functional neuroimaging abnormalities associated with IGE. All patients were diagnosed with IGE based on ILAE criteria[2], including different IGE subtypes: eight studies included IGE patients with generalized tonic-

clonic seizures (GTCS), seven with juvenile myoclonic epilepsy (JME), two with absence epilepsy (AE), and four with mixed IGE subtypes. Coordinates of gray matter atrophy using voxel-based morphometry were reported in 13 studies and coordinates of increased spontaneous local activity using resting-state functional MRI ("fMRI hyperactivity") were reported in eight studies (Supplementary Tables 1–3). Coordinates of neuroimaging abnormalities (21 studies, 131 coordinates) were heterogeneously distributed across the brain and involved different cortical lobes, thalamus, basal ganglia, hippocampus, brainstem, and cerebellum (Fig. 1A–C).

### Activation likelihood estimation (ALE) meta-analysis

ALE meta-analysis of the 131 coordinates from all 21 studies identified the bilateral thalamus (anterior, mediodorsal and ventral posterior lateral nuclei) as regions consistently implicated across studies (Fig. 1B and Supplementary Table 4). Repeating the ALE meta-analysis separately for coordinates of brain atrophy or fMRI hyperactivity identified different regions within the thalamus and cerebellum without any overlap (Supplementary Fig. 2). Notably, only 17% of all coordinates were in the thalamus (Fig. 1C).

### Coordinate network mapping

We performed coordinate network mapping using the human connectome and found that these heterogeneously distributed coordinates were connected to a common brain network. This network consists of positive connectivity to the supplementary motor area (SMA), sensorimotor cortex (pre- and post-central gyri), superior temporal gyrus, anterior cingulate, piriform cortex, putamen, centromedian thalamus, and cerebellum (peak positive overlap of >85% in the centromedian thalamus, cerebellum, putamen, and piriform cortex); and negative connectivity to the medial frontal lobe, parieto-occipital, precuneus, middle and inferior temporal gyri (peak negative overlap of >85% in the frontal poles, middle frontal gyrus, angular gyrus, and precuneus). Hereafter, we will refer to this network (i.e., the overlap of all 21 study-level coordinate networks) as an "IGE network" (Fig. 2B). This network was specific to IGE compared to coordinates from neuroimaging abnormalities in neurodegenerative disease ($P_{FDR} < 0.05$, Fig. 2D) and random coordinates ($P_{FDR} < 0.05$, Fig. 2E).

The identified IGE network showed a consistent topography across many different variations in the methods. First, we repeated the

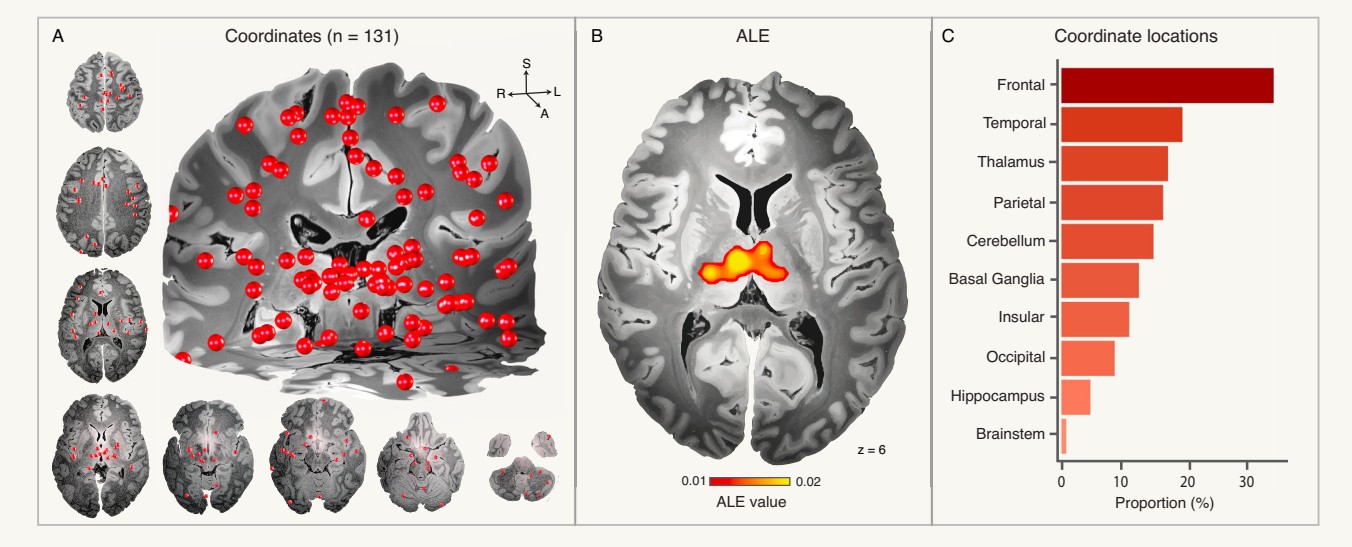

**Fig. 1 | Coordinate locations and ALE.** Coordinates (*n* = 131), shown as red spheres, were heterogeneously distributed across the brain (**A**). An ALE analysis identified consistent brain abnormalities in the bilateral thalamus (**B**). Notably, only 17% of all coordinates were in the thalamus (**C**). *Note that spheres of coordinates may be assigned to multiple lobes or regions when located at the borders between them. Source data are provided as a Source Data file. ALE activation likelihood estimation.

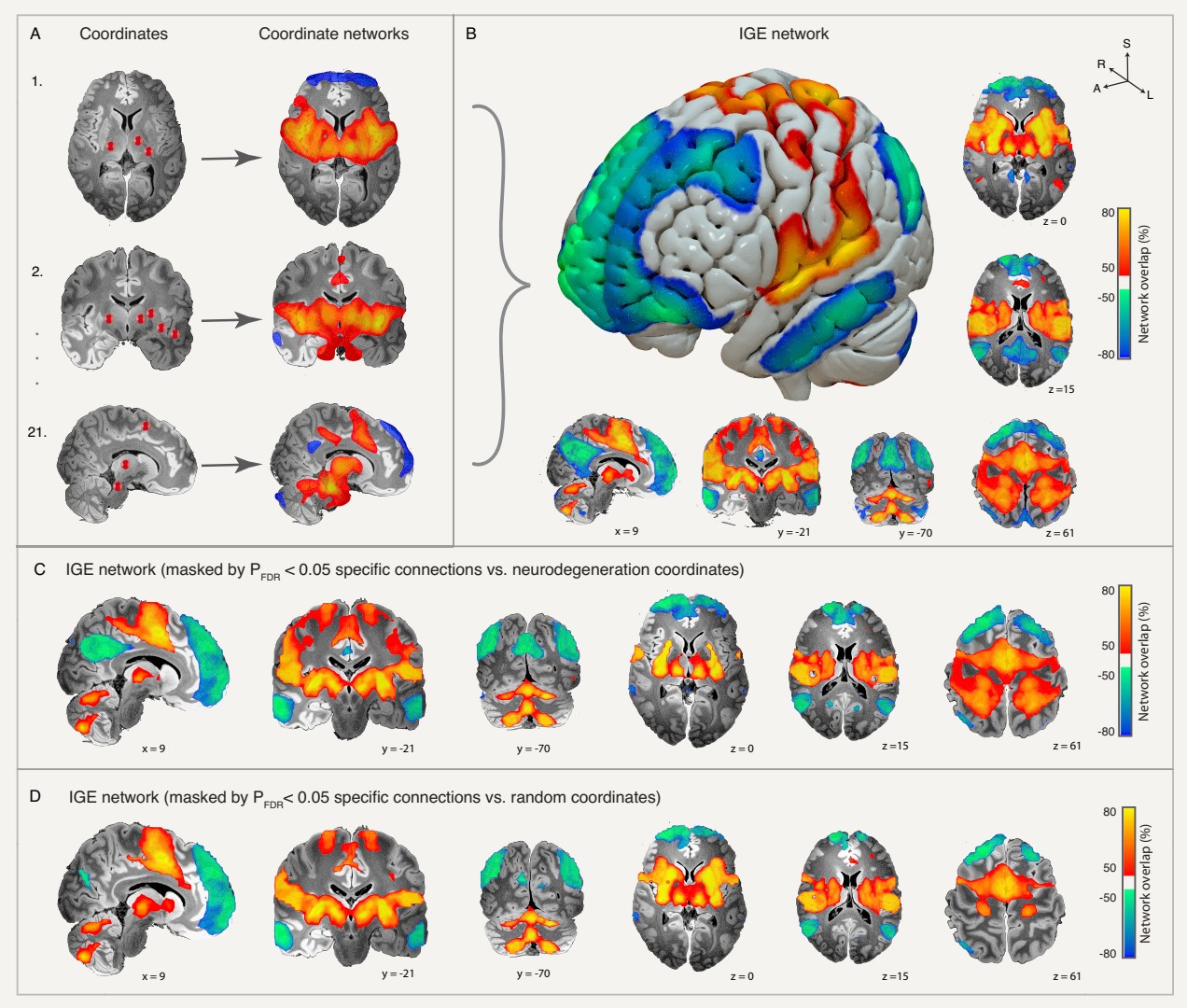

**Fig. 2 | Coordinate network mapping.** Study-level coordinates (**A**, left) were used as seeds in an atlas of human brain connectivity (i.e., a human connectome) to identify the functional brain network connected to these coordinates. (**A**, right). Coordinate networks for each study (*n* = 21) were then overlapped to identify a common brain network deriving an IGE network (**B**). These same functional connections were found to be specific to IGE compared to coordinates from neuroimaging abnormalities in neurodegenerative diseases (**C**) or randomly distributed coordinates (**D**). FDR false discovery rate, IGE idiopathic generalized epilepsy.

coordinate network mapping analysis using independent normative adult or pediatric connectomes, and a disease-specific connectome derived from IGE patients, each of which identified a similar IGE network (Supplementary Fig. 3). Second, we found that results were not dependent on coordinate size (Supplementary Fig. 4) or arbitrary thresholds of our study-level coordinate networks (Supplementary Fig. 5). Third, repeating the analyses separately for studies of gray matter atrophy or fMRI hyperactivity identified a similar network (Supplementary Fig. 6) as did using each coordinate as a seed instead of using study-level seeds (Supplementary Fig. 7). Fourth, using a leave-one-diagnosis-out analysis, we found that this IGE network was not driven by any one predominant IGE subtype (Supplementary Fig. 8). Coordinate network mapping analyses of each IGE subtype showed similar subcortical connectivity profiles across different diagnoses, but slightly different cortical connectivity profiles. Notably, GTCS and JME subtypes were positively connected to the motor cortex, while the AE subtype was negatively connected (Supplementary Fig. 9).

## Multimodal validation

The worldwide ENIGMA study identified significant atrophy in the bilateral precentral gyri and the right thalamus[17]. These same brain regions were part of the identified IGE network and overlapped more with the network compared to a null distribution of regions randomly selected from the same atlas ($t = 0.53$, $P = 0.015$, Fig. 3A). To test whether the identified IGE network may help explain the brain areas activated during generalized-onset epileptiform discharges, we identified 12 studies performing simultaneous (inter)ictal EEG-fMRI in patients with IGE or focal epilepsies and extracted the coordinates of fMRI activation at the time of discharge. Brain regions activated during simultaneous (inter)ictal EEG-fMRI in patients with IGE aligned more with the identified IGE network compared to regions activated in patients with focal epilepsy ($t = 2.96$, $P = 0.015$, Fig. 3B), and aligned more with the IGE network than a coordinate network map for neurodegenerative diseases ($t = 8.34$, $P = 0.0004$, Supplementary Fig. 10). We investigated alignment between the IGE network and a previously identified brain network derived from brain lesions associated with focal epilepsy[30] and found they were inversely related ($r = -0.470$). Lesions associated with epilepsy overlapped with negative functional connections in the IGE network, while lesions not associated with epilepsy overlapped with positive functional connections ($t = 11.05$, $P < 0.001$, Fig. 3C). IGE network alignment with scalp EEG was investigated using average sampling locations of scalp EEG electrodes

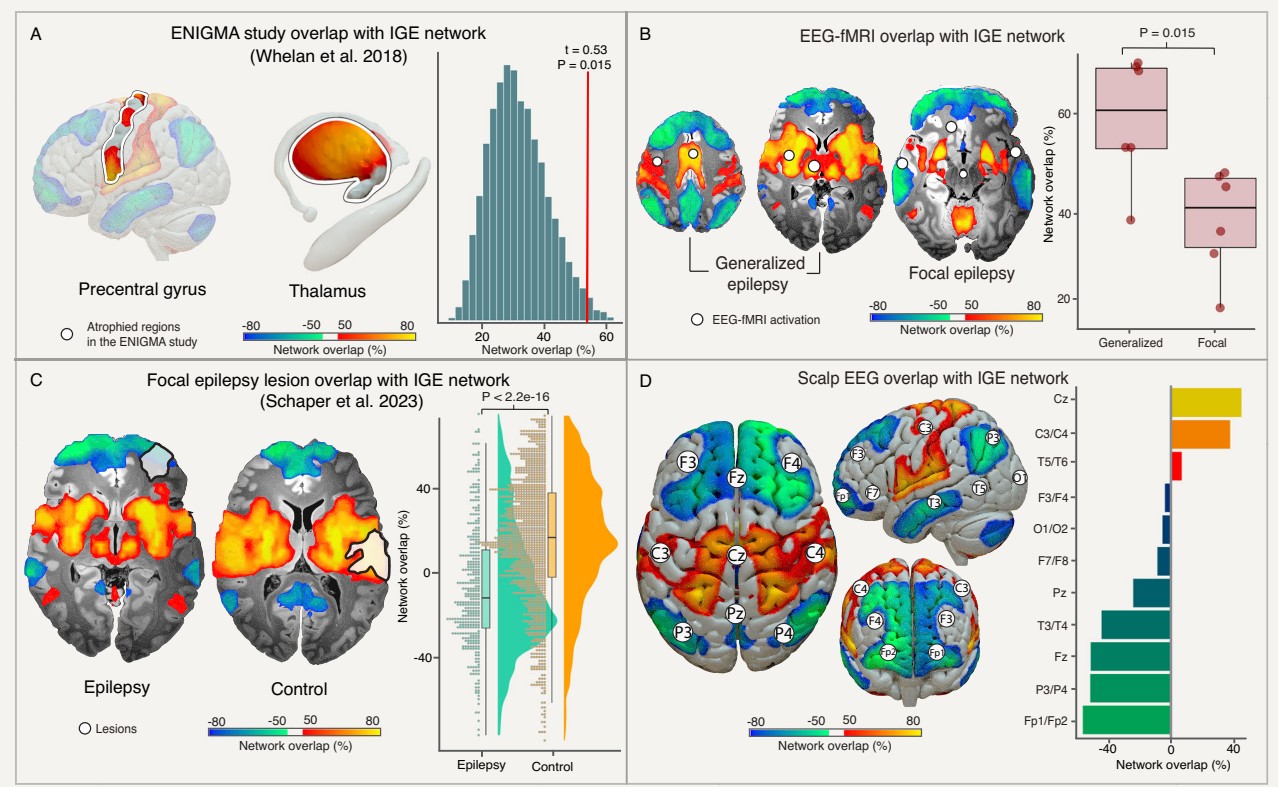

**Fig. 3 | Multimodal validation.** Brain regions atrophied in IGE (white outlines) as identified in the worldwide ENIGMA study[17] were part of the IGE network and overlapped more with this network compared to a null distribution of randomly selected brain regions (**A**). Brain regions activated by epileptiform discharges during simultaneous EEG-fMRI (white dots) in generalized epilepsies (n = 6) show higher overlap with the IGE network compared to focal epilepsies (n = 6) in a two-sided t-test (t = 2.955, df = 9.7124, P = 0.0149, 95% CI [0.038, 0.389]). This is shown in the boxplot where the center is the median overlap with the IGE network bound by the 25th and 75th percentile of the data. The whiskers extend from the lower and upper quartile to the minimum and maximum, respectively. The points on the plot each mark a unique EEG-fMRI study on either generalized or focal epilepsy (**B**). Lesion locations[30] associated with epilepsy (n = 347) overlap with

negative functional connections in the IGE network more than lesions not associated with epilepsy (n = 1126) in a two-sided t-test (t = 12.69, df = 581.9, P < 2e-16, 95% CI [0.2,+∞]). The distribution of the data is shown in a raincloud plot which displays individual epilepsy and control cases in the jitter plot. The boxplot shows the median network overlap of these two groups by the center line and is bound by the 25th and 75th percentiles of the data. The whiskers of the plot mark data that fall within 1.5 times the interquartile range (**C**). Scalp EEG electrodes (white circles) in frontocentral regions (Cz) overlap with positive functional connections of the IGE network, while electrodes in anterior frontal regions (Fp1/Fp2) overlap with negative functional connections (**D**). Source data are provided as a Source Data file. EEG-fMRI electroencephalogram-functional magnetic resonance imaging, IGE idiopathic generalized epilepsy.

according to the international 10–20 system[41]. Scalp EEG electrodes in frontocentral regions (Cz and C3/C4) aligned with positive functional connections, while electrodes in frontopolar (Fp1/2) and posterior regions (P3/4) aligned with negative functional connections (Fig. 3D).

## Alignment with the somato-cognitive action network

The IGE network includes brain regions previously implicated in the control of movement and shows a discontinuous pattern over the motor cortex, reminiscent of the recently identified inter-effector regions[42]. We tested alignment between the coordinates of these inter-effector regions and the identified IGE network. The inter-effector regions aligned more with the IGE network compared to the coordinates of the leg, hand, and mouth effector regions (t = 3.96, P = 0.005, Fig. 4A–C). A similar result (P < 0.0001, Supplementary Fig. 11) was found in computing the connectivity between the study-level IGE coordinates and the (inter)effector regions in each of the subjects of the human connectome. The IGE network shared connections with the somato-cognitive action network (SCAN, spatial r = 0.81, Fig. 4D, E), and study-level IGE coordinates were most positively connected to the SCAN compared to any other canonical brain network, but most negatively connected to the default mode network (DMN, one-way ANOVA $F_{(11,651)}$ = 1989, P < 0.0001, Fig. 4F).

## Relevance to deep brain stimulation

The IGE network involved peak functional connectivity to the centromedian nucleus (CM) of the thalamus, a region that has been used as a DBS target to treat generalized seizures[43] (Fig. 5A–C). We analyzed the DBS electrode locations and clinical outcome (% reduction in seizure frequency) of 21 patients with IGE (15 females, 5 males, 1 non-binary) that were treated with CM DBS for drug-resistant generalized seizures. Seizure frequency reduced a median of 90% (interquartile range: 66.5–96.5%), 66.7% on average (standard error: 14.3) and 19 of 21 patients (90%) were responders with >50% reduction in seizure frequency considered clinically significant (Fig. 5D). DBS electrode locations aligned with the topography of the IGE network in the thalamus (Fig. 5E). Notably, the IGE network peak in the thalamic CM (Fig. 6A) converged in the same location (MNI coordinate: x = −9.05, y = −21.07, z = −0.07) as a recently identified optimal DBS site ("sweetspot") for IGE[44] (Fig. 6B), and 4 mm closer compared to a DBS sweetspot for patients with Lennox−Gastaut syndrome (LGS)[45] (Fig. 6C). To illustrate the clinical potential of these findings to inform image-guided DBS for generalized epilepsy, we localized the DBS electrodes of an independent patient (1 female) with IGE treated in our center at the Brigham and Women's Hospital and visualized this IGE network peak, as well as the previously reported DBS sweetspots in patients with IGE or LGS, and

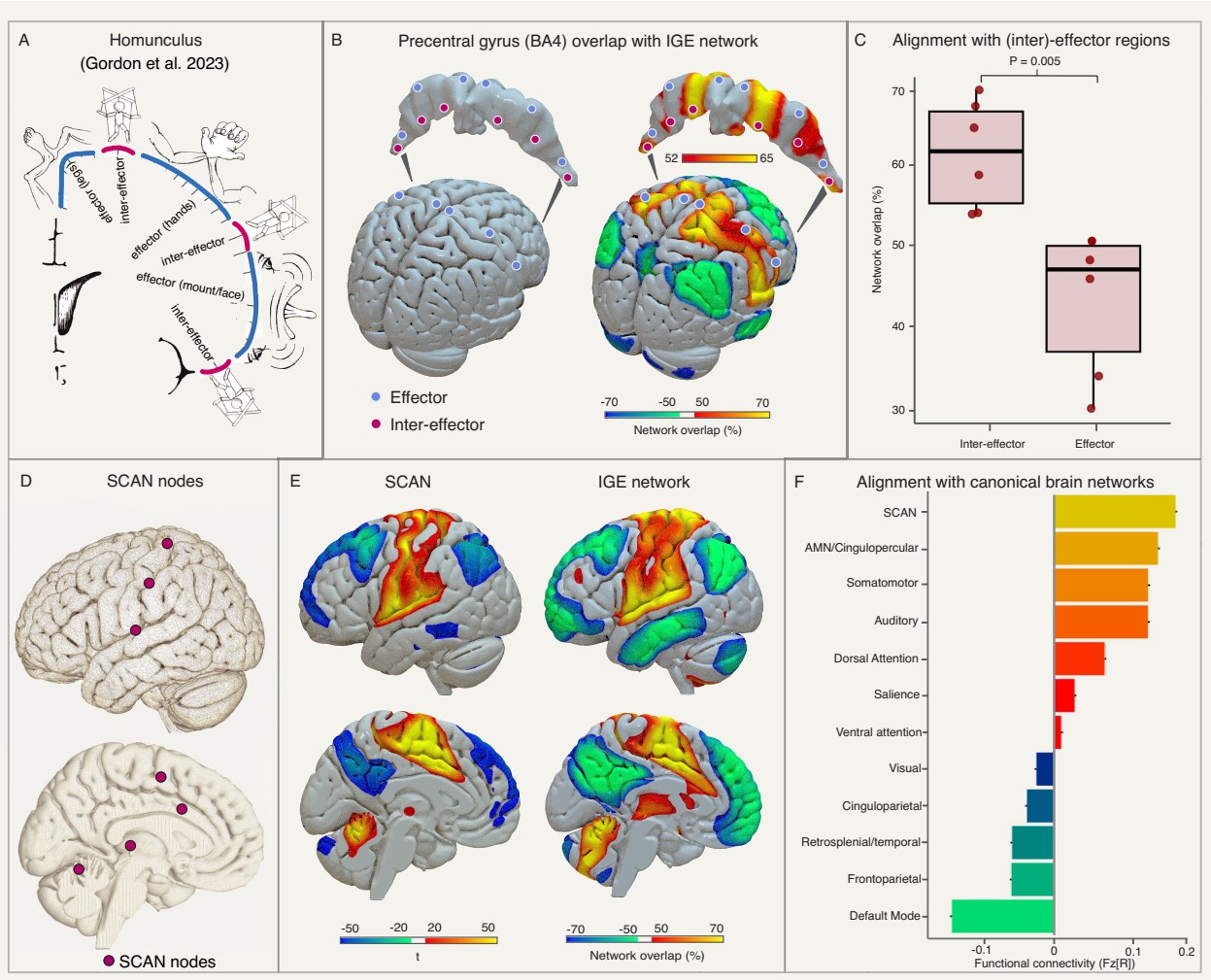

**Fig. 4 | Alignment with the somato-cognitive action network.** The motor cortex homunculus (**A**, panel adapted under the Creative Commons Attribution (CC-BY) license from Gordon et al. 2023[42]) includes effector and inter-effector regions (**B**, left)[42]. Inter-effector regions (n = 6) align with the IGE network (**B**, right) and show higher overlap compared to effector regions (*n* = 5) in a two-sided *t*-test (*t* = 3.96, df = 7.17, *P* = 0.005, 95% CI[0.081, 0.38]). The boxplot shows the center line as the median network overlaps for each effector and inter-effector region (dots) enclosed by the 25th and 75th percentiles of the data, while the whiskers extend to the maximum and minimum (**C**). The peak nodes in the SCAN (**D**) were used as a seed in the human connectome to generate a whole-brain map of the SCAN (**E**, left), which was spatially similar to the identified IGE network (spatial *r* = 0.81; **E**, right). Coordinates of brain abnormalities in IGE were most positively connected to the SCAN and most negatively connected to the default mode network (**F**). Source data are provided as a Source Data file. IGE idiopathic generalized epilepsy, SCAN somato-cognitive action network.

discriminative fibers of the reticular system (Fig. 6D) associated with improved generalized seizure control after CM-DBS.

### A convergent generalized epilepsy network

The IGE network (Fig. 7A) derived from coordinate networks of brain abnormalities was spatially similar (spatial *r* = 0.672) to a CM DBS network (Fig. 7B) derived from the networks of the individual patient's DBS sites weighted by clinical outcome (% reduction in seizure frequency). A convergent IGE network was identified by averaging the IGE network and CM DBS network (spatial *r* to IGE network = 0.914; spatial *r* to CM DBS network = 0.914, Fig. 7C).

### Discussion

In this study, we identified a generalized epilepsy network by combining brain abnormalities and DBS data with an atlas of human brain connectivity (i.e., a human connectome). There are four key findings. First, coordinates of heterogeneously distributed neuroimaging abnormalities in patients with IGE were connected to a common brain network. Second, this network included structural abnormalities in IGE previously identified in the worldwide ENIGMA study and brain areas activated by generalized epileptiform discharges in simultaneous EEG-fMRI studies. Third, the network aligned with the inter-effector regions of the motor cortex and shared brain network topography with the recently identified SCAN. Fourth, the IGE network peaked in the CM nucleus of the thalamus, a DBS target associated with a median 90% reduction in seizure frequency in patients with IGE. These findings could be relevant for our understanding of generalized epilepsy as a network disease, help explain seizure semiology, or identify therapeutic targets for brain stimulationt.

The ALE meta-analysis of coordinate locations identified the anterior, mediodorsal, and ventral posterolateral thalamus as the brain regions most consistently implicated across both structural and functional MRI studies of IGE. This finding is consistent with a previous ALE meta-analysis of structural brain abnormalities in IGE[25], the worldwide ENIGMA study, which identified thalamic volume loss in IGE[6,17] and thalamic activation in simultaneous EEG-fMRI studies[46,47]. However, only 17% of the total number of coordinates from our

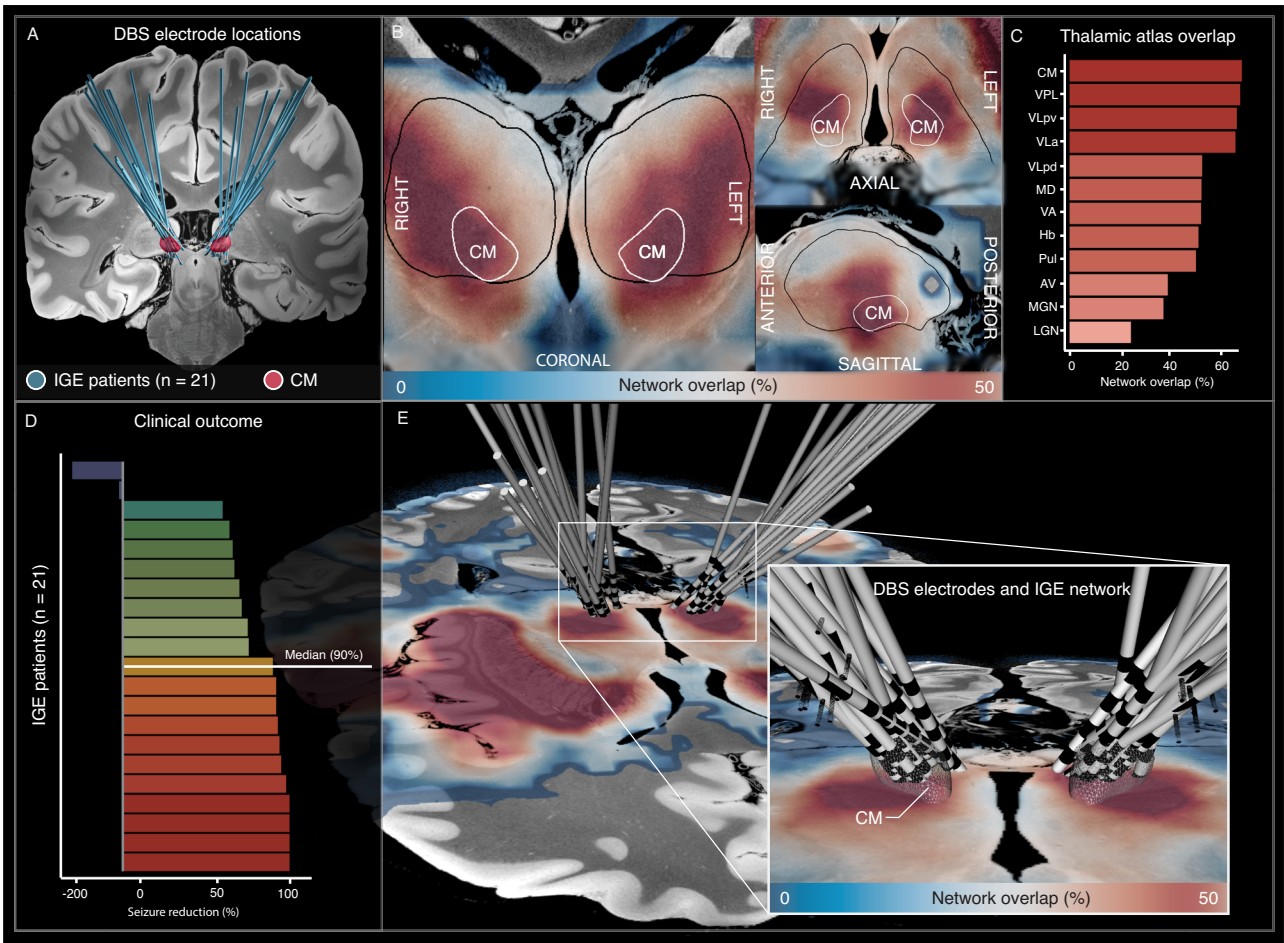

**Fig. 5 | Relevance for deep brain stimulation.** DBS electrode locations implanted to treat drug-resistant generalized seizures in patients with IGE (*n* = 21) were localized with Lead-DBS software[118] and plotted in relation to the CM (**A**, red). Notably, the IGE network peaked in the CM of the thalamus (**B**), which was the most functionally connected thalamic nucleus[119] (**C**). Seizures reduced a median 90% after CM-DBS in 21 patients with IGE (**D**). The IGE network was projected onto a publicly available ultra-high resolution ex vivo brain aligned to MNI space[117] (**E**, warm colors), and DBS electrodes intersected with the peak of the IGE network in the CM

(mesh). Source data are provided as a Source Data file. AV anterior ventral nucleus, CM centromedian nucleus, DBS deep brain stimulation, Hb habenular nucleus, IGE idiopathic generalized epilepsy, LGN lateral geniculate nucleus, MD mediodorsal nucleus, MGN medial geniculate nucleus, Pul pulvinar nucleus, SCAN somato-cognitive action network, VA ventral anterior nucleus, VLa ventral lateral anterior nucleus, VLpd ventral lateral posterior nucleus (dorsal part), VLpv ventral lateral posterior nucleus (ventral part), VPL ventral posterior lateral nucleus.

---

systematic search were in the thalamus, suggesting involvement of a wider brain network.

Using the human connectome and coordinate network mapping, we found that these same coordinates mapped to a common brain network despite being heterogeneously distributed. This finding is consistent with an increased understanding of IGE as a brain network disease[5,6,9] and previous coordinate network mapping studies[27,34,35,39]. The IGE network comprised cortical and subcortical areas, including positive functional connectivity to SMA, sensorimotor cortex, anterior cingulate, superior temporal gyrus, piriform cortex, thalamus, basal ganglia, and cerebellum; and negative functional connectivity ("anticorrelation"[48,49]) to the frontal poles, medial frontal lobe, angular gyrus, precuneus, middle and inferior temporal gyri. These regions have previously been implicated in the generation of spike and wave discharges in animal models of generalized epilepsy[50,51], align with structural abnormalities in IGE previously identified in the worldwide ENIGMA study[17] and results from other surface-based neuroimaging studies in patients with IGE[6,7,10,17,18,52,53].

Our findings may also help unify prior neuroimaging and EEG findings in IGE. Brain areas activated by generalized epileptiform discharges in simultaneous EEG-fMRI studies, such as the sensorimotor

cortex and thalamus, align with the positive functional connections of the IGE network[54–57]. Yet, routine scalp EEG in IGE patients typically shows a frontal predominance of GSW's in anterior and medial frontal scalp EEG electrodes[56,58,59], which aligns with the negative functional connections of the IGE network. Some source imaging studies likewise find an anteriorly predominant frontal GSW generator in the medial prefrontal and superior frontal cortex[60–63], although results have varied according to the timing of source analysis in relation to discharge onset[63,64]. In contrast to the sensorimotor cortex and thalamus (positive functional connections), anteromedial frontal regions (negative functional connections) are typically deactivated in EEG-fMRI studies of IGE[46,65–69]. The identified IGE network thus suggests an opposing functional relationship between the brain regions atrophied and activated in IGE versus the brain regions of a potential GSW source.

One interpretation of this finding may be that atrophy reflects changes in a network attempting to inhibit or compensate for seizures, consistent with previous network mapping findings[30,39,70,71] This could potentially be supported by different components of the GSW discharge localizing to different brain regions, which is a testable hypothesis for future work. While more data is needed, spikes of the spike-wave complex are usually distributed over anterior and medial

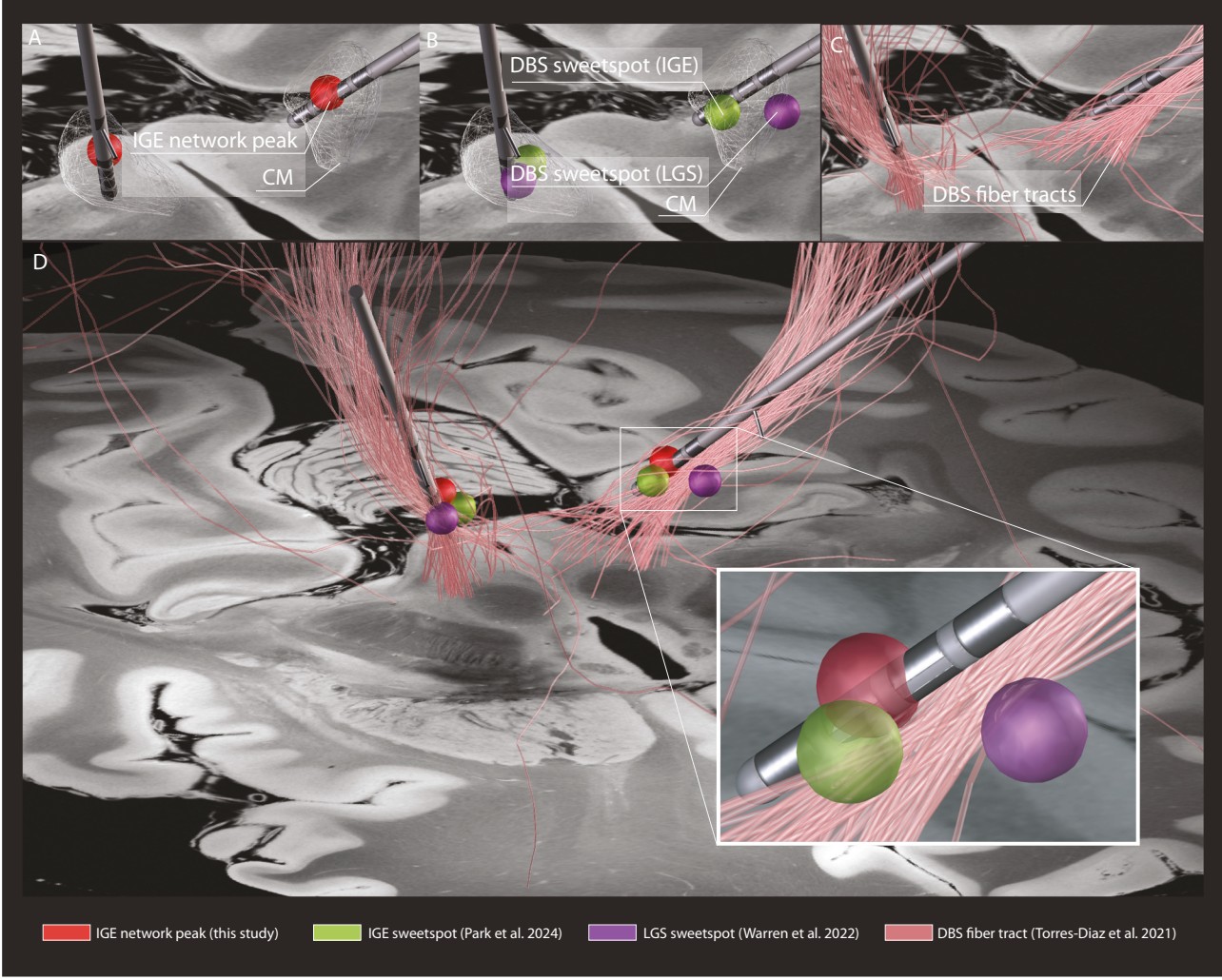

**Fig. 6 | Illustration of the potential clinical translation to image-guided DBS.** The DBS electrode locations of an independent patient with IGE treated with CM DBS was plotted in relation to (1) the peak voxel of the IGE network (MNI coordinate: x = −9.05, y = −21.07, z = −0.07) in the thalamus (**A**, red), (2) previously reported DBS sweetspots in IGE[44] (**B**, green) and LGS[45] (**B**, purple), and (3) discriminative fiber tracts[91] associated with improved generalized seizure control after CM DBS (**C**, pink). This IGE network peak converged on a similar location to these optimal DBS sites, yet 4 mm closer to the sweetspot derived from IGE versus LGS patients (**D**). CM the centromedian nucleus, DBS deep brain stimulation, IGE idiopathic generalized epilepsy, LGS Lennox–Gastaut Syndrome.

frontal regions (negative functional connections), yet waves are frequently distributed over central regions (positive functional connections)[72,73]. This is in line with the opposite direction of functional connectivity in the IGE network identified here and the hypothesized opposite physiological roles of spikes ('excitation') and waves ('inhibition') during thalamocortical oscillations[74,75]. Finally, routine scalp EEG typically shows GSWs, but focal EEG features are seen in approximately 30% of patients with IGE. Focal features in IGE predominantly localize to the temporal lobe[72], consistent with the topography of negative functional connections of the IGE network and a brain network derived from lesions associated with focal epilepsy[30]. This IGE network may, therefore, provide a potential mechanism to explain generalized epilepsy with clinical and electrographic focal features[76,77]. Overall, these network results could help unify previous neuroimaging and EEG findings in IGE and are consistent with the broader notion that generalized epileptiform discharges reflect recruitment of a specific bilaterally distributed, large-scale brain network, rather than the 'whole brain'[75,78,79].

Our study presents the spatial topography of a human brain network implicated in IGE. The identified IGE network included positive functional connectivity to the sensorimotor cortex. This finding is

consistent with GSW-related fMRI activation in simultaneous EEG-fMRI studies[54–56]. Seizure-related activation or disruption of these regions could help explain the typical symptoms seen in generalized tonic-clonic seizures and myoclonic jerks. More specifically, we observed a discontinuous pattern over the motor cortex that aligned with the inter-effector regions of the revised motor cortex architecture and a network topography reminiscent of the SCAN[42]. The SCAN is hypothesized to form part of an integrated action and executive control system to coordinate gross movements, control muscle groups, posture, and internal physiology[42]. This finding may help explain why generalized tonic-clonic seizures typically show convulsions of the whole body rather than any single arm or leg, electrographically start or quickly spread widely across the brain[59,80], and are often associated with autonomic changes[81]. Accordingly, seizure activity may "hijack" (parts of) the SCAN resulting in the electroclinical expression of generalized seizures along intrinsic human brain networks[80], akin to the secondary epilepsy network hypothesis of LGS[82]. While brain abnormalities in IGE were most positively connected to the SCAN, they were most anticorrelated to the DMN. This finding is consistent with previous EEG-fMRI studies showing fMRI de-activation throughout the DMN during GSW[56,58,59] and loss of consciousness typically seen with

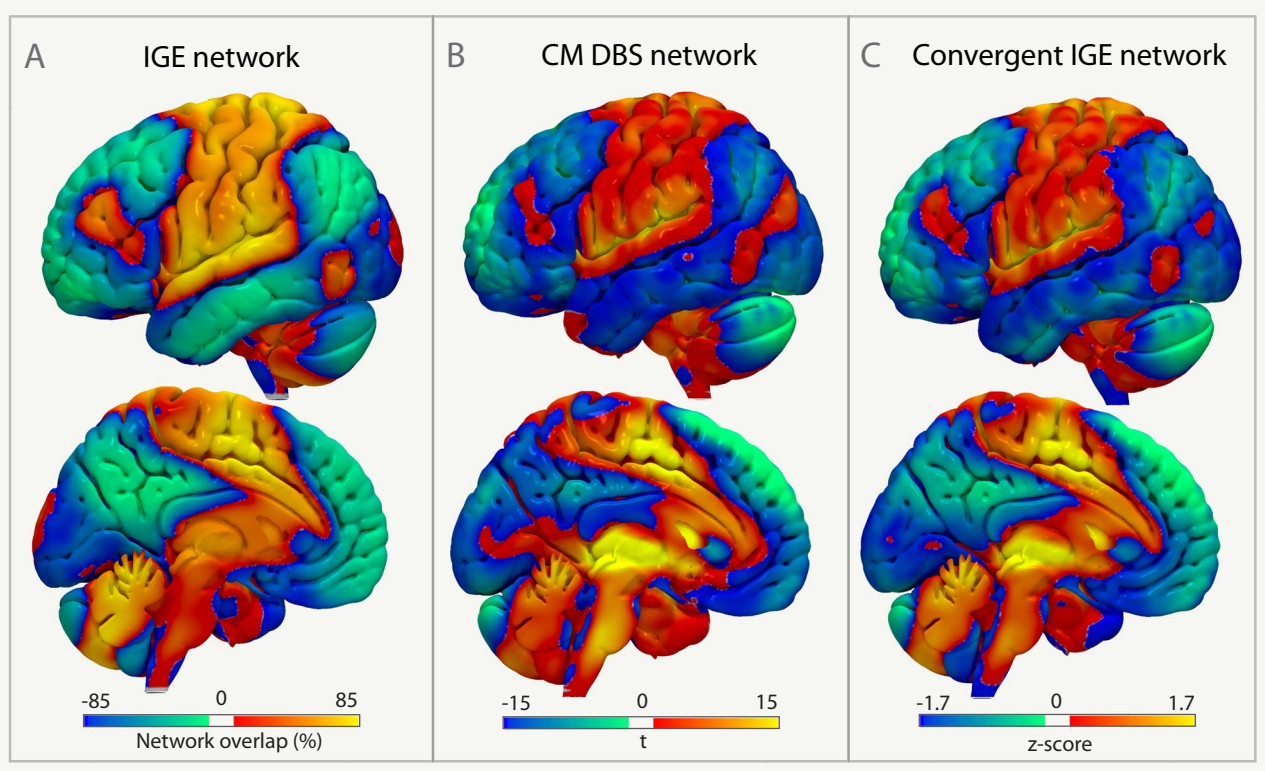

**Fig. 7 | A convergent generalized epilepsy network.** The IGE network (**A**) was spatially similar to the CM DBS network (**B**), and a convergent IGE network was identified (**C**). CM the centromedian nucleus, DBS deep brain stimulation, IGE idiopathic generalized epilepsy.

generalized tonic-clonic[47] or absence seizures[80]. In a separate analysis of each IGE subtype, the AE subtype showed a similar subcortical connectivity profile compared to the GTCS and JME subtypes, but the cortical connectivity profile differed slightly. GTCS and JME coordinates were positively connected to the motor cortex, while AE coordinates were anticorrelated to the motor cortex, consistent with the lack of movements during absence seizures. Involvement of the motor cortex may thus be subtype-specific suggesting IGE and its subtypes may involve both shared and subtype-specific (sub)cortical brain networks[83,84].

Our findings may also have therapeutic implications for brain stimulation treatment in generalized epilepsy. Specifically, they suggest that the CM nucleus in the thalamus is a key node in an IGE network consistent with recordings of epileptiform discharges in the CM region during generalized seizures[85–87] and early involvement of the CM during GSW discharges in EEG-fMRI[88]. Our IGE network results may help explain the positive results of CM DBS and RNS in patients with IGE as reported by case- and open-label studies[89,90]. Brain networks connected to neuroimaging abnormalities in IGE thus converge on a DBS target used to treat generalized seizures[45,91]. Serendipitous, the IGE network peaked in the same location in the thalamus as an optimal DBS site ("sweetspot") for patients with IGE[44]. Furthermore, a CM DBS network derived from the stimulation sites of patients with IGE recapitulated the IGE network. Overall, these findings suggest potential clinical utility of the IGE network to identify brain stimulation targets and could be used to guide DBS, responsive neurostimulation (RNS) or non-invasive brain stimulation therapies such as transcranial magnetic stimulation, multifocal transcranial electric stimulation, and focused ultrasound[92].

Strengths of our study include a systematic search and meta-analysis of the published literature on voxel-based neuroimaging abnormalities in IGE; consistency of results across multiple variations in the methods, including different adult and pediatric normative

connectomes and a disease-specific IGE connectome; leave-one-diagnosis-out-analyses; and multimodal support of network localization using different neuroimaging modalities and DBS data. There are several limitations to consider. First, while this study highlights shared network connections across IGE subtypes, this does not preclude potentially important differences between subtypes. Future studies using single-subject level data in combination with modern volume- and surface-based analysis methods could more precisely identify structural abnormalities in patients. Such efforts may help identify specific networks involved in different IGE subtypes, seizure types, or even patient-specific seizure networks[93,94]. Second, it remains unclear whether brain abnormalities such as atrophy in IGE are a cause, consequence, or compensatory mechanism of epilepsy. Several interpretations could be made. They could reflect changes in a network damaged by seizures and thus be a direct consequence of seizures. They could also reflect abnormal brain development that is causing seizures. Other hypotheses suggest atrophy is part of a degenerative process that facilitates epilepsy, an effect of antiseizure drugs, or some combination of these factors[84]. A different potential interpretation may be that atrophy reflects changes in a network attempting to inhibit or compensate for seizures[30,71]. The finding that brain lesions associated with focal epilepsy are anticorrelated to the brain regions atrophied in IGE suggests they involve different brain networks and that atrophy may be a consequence or compensatory mechanism in epilepsy rather than a cause, consistent with previous network mapping findings[30,39,70,71]. Future prospective longitudinal studies are needed to elucidate the causal cascade of these structural network changes in relationship to clinical outcomes and the spatiotemporal dynamics of seizures within this network. Third, due to the small number of subjects and almost uniformly high response rate (90%) after CM DBS in patients with IGE, we were underpowered to detect connections that covary with clinical response. Future work in larger cohorts could revisit this issue, and a clinical trial of RNS of the CM in

patients with IGE is ongoing (ClinicalTrials.gov: NCT05147571). Fourth, as is inherent to meta-analyses of published group-level data, we did not have access to detailed clinical phenotyping such as seizure frequency or severity, cognitive or mood comorbidities, or antiseizure drug use[95]. However, these variables should increase heterogeneity in the studied population and add noise, biasing us against identifying convergent results across studies. Fifth, all the coordinates utilized in this study were derived from gray matter. Given that fMRI signals in white matter are now acknowledged as more than mere noise[96–98], future research should incorporate findings from both white and gray matter to map their shared networks[99]. Finally, our study was based on retrospective analyses of existing data, and any clinical implications should thus be interpreted with caution. Future prospective studies are needed to determine if this network can be used as a safe an effective brain stimulation target.

In summary, we identified a generalized epilepsy network that links heterogeneously distributed brain abnormalities in IGE to a common brain network and DBS sites reducing generalized seizures. This generalized epilepsy network could help guide future clinical trials of brain stimulation to better control generalized seizures.

## Methods

This study was carried out in accordance with the Declaration of Helsinki, approved by the institutional review board of the Brigham and Women's Hospital, Boston, Massachusetts. The coordinate network mapping analysis was exempted from obtaining informed consent based on the secondary use of published data. Any patient data used in this study was obtained with informed consent, including secondary use of research data. Preferred reporting items for systematic reviews and meta-analyses (PRISMA) guidelines were followed to identify published coordinates of neuroimaging abnormalities associated with IGE.

### Systematic search and coordinates

In line with best-practice recommendations for coordinate-based meta-analysis[100], we systematically searched the literature for structural and functional MRI (fMRI) studies in IGE patients compared to healthy controls. We included studies reporting coordinates of gray matter atrophy using voxel-based morphometry and studies reporting increased spontaneous local activity using resting-state fMRI. The following search terms were used in PubMed and EMBASE databases: "voxel-based morphometry", "resting-state functional MRI", "ALFF", "ReHo", "epilepsy", and their derivatives (Supplementary Table 1). All reported coordinates of gray matter atrophy or increased spontaneous local activity ("fMRI hyperactivity") in IGE patients compared to healthy controls were extracted from the published studies (Supplementary Tables 2, 3). Coordinates were recorded in the Montreal Neurological Institute (MNI)-ICBM-152 space. A detailed description of the search strategy and study selection can be found in Supplementary Methods 1.

### Activation likelihood estimation (ALE) meta-analysis

We performed a standard coordinate-based meta-analysis using activation likelihood estimation (ALE) with GingerALE software (Version 3.0.2) to identify brain areas that were consistently implicated across studies compared to a null distribution of 10,000 randomly distributed coordinates[101]. Family-wise error [FWE] rate correction for multiple testing was performed at the cluster level ($P_{FWE} < 0.01$, with a cluster-defined threshold of $P < 0.001$, 10,000 permutations), in line with published recommendations[102].

### Human connectome

To identify the functional connections of these coordinate locations, we used a normative functional connectome derived from the resting-state fMRI data of 652 healthy Asian adults (mean age ± standard

error = 22.9 ± 5.53 years old, 334 females)[94]. All participants provided written informed consent. Acquisition and preprocessing of the fMRI data were described previously[103,104], and is consistent with pre-processing of the Brain Genomics Superstruct Project (GSP, https://dataverse.harvard.edu/dataverse/GSP)[105] normative connectome. Detailed steps can be found in Supplementary Methods 2[106].

### Coordinate network mapping

We performed coordinate network mapping according to previously described methods[27,34]. In short, we first create a spherical seed (6 mm diameter) centered at each reported coordinate. For studies reporting multiple coordinates, coordinates were combined into one volume or study-level seed, as different coordinates from one study are not independent. Seed-to-whole-brain functional connectivity was computed on the preprocessed rs-fMRI data of 652 healthy subjects[94] using Pearson correlations. As in prior studies[27–29], a one-sample t-test was used to identify the voxels that were significantly connected to each seed (i.e., showing correlated fMRI signal fluctuations), resulting in a coordinate network. Each coordinate network was then thresholded and binarized ($-5.1 < t > 5.1$; voxel-wise FWE correction, $P < 0.01$) as previously reported. The resulting coordinate networks were over-lapped (i.e., summed and divided by the number of studies) to identify the common network connections across all studies, also termed a coordinate network overlap map. The value of each voxel within this coordinate network overlap map indicates the proportion of studies with coordinates functionally connected to that voxel. Henceforth, we refer to this unthresholded coordinate network overlap map as the 'IGE network'.

We recreated the IGE network using many different variations in the methods to assess the consistency of our results. First, we used different independent normative connectomes including a Western adult connectome derived from the GSP ($n = 1000$ GSP connectome)[105], and pediatric (9–10-year old) connectome derived from the adolescent brain cognitive development study ($n = 1000$ ABCD connectome)[107,108]. Second, as IGE may be associated with changes in functional connectivity[79,109], we recreated the IGE network using a disease-specific connectome derived from patients with IGE ($n = 172$, Supplementary Methods 3). Third, study-level coordinates were recreated using different diameters (3 mm and 9 mm) to assess the potential influence of coordinate size. Fourth, study-level coordinate networks were binarized using different thresholds (positive and negative $t = 4.7, 5.6, 7,$ and 9) to assess the potential influence of arbitrary statistical thresholds to create the coordinate networks[110]. Fifth, we created separate coordinate network overlap maps for study-level coordinates derived from structural or functional neuroimaging abnormalities. Sixth, we recreated the IGE network using the individual coordinates as seeds as opposed to the study-level coordinates. Seventh, we performed a leave-one-diagnosis-out analysis to assess whether our results were driven by a particular subtype such as IGE with generalized tonic-clonic seizures (GTCS), juvenile myoclonic epilepsy (JME), or absence epilepsy (AE). The coordinate network mapping analysis was also repeated for each IGE subtype in a separate analysis. Spatial similarity of the IGE network across multiple sensitivity and control analyses was computed using a spatial correlation (Pearson's r).

### Specificity testing

To determine whether the IGE network was specific to neuroimaging abnormalities in IGE and not a result of nonspecific atrophy or our choice of connectome, we compared the IGE coordinate networks to control networks derived from previously published coordinates of atrophy in neurodegenerative diseases[27] and randomly distributed coordinates. The neurodegenerative disease coordinate networks included 49 study-level networks from a previous study (including Alzheimer's disease, $n = 8$; behavioral variant frontotemporal

dementia, $n = 21$; corticobasal syndrome, $n = 12$; and progressive non-fluent aphasia, $n = 8$, Supplementary Table 5)[27]. A null distribution of randomly distributed coordinates were created by redistributing the study-level coordinates of the IGE studies at random within the brain. This was performed four times resulting in 84 randomly distributed study-level coordinates. Voxel-wise two-sample $t$-tests were performed using the software permutation analysis of linear models (PALM) in FSL (V6.0.4), correcting for multiple comparisons using threshold-free cluster enhancement and an FDR-corrected $P < 0.05$ was considered significant.

## Multimodal validation

We validated and investigated the topography of the IGE network using a multimodal approach. We compared the IGE network to findings in previous studies by testing IGE network overlap of: (1) structural abnormalities in IGE identified in the worldwide ENIGMA study, (2) brain regions activated during simultaneous (inter)ictal EEG-fMRI of generalized spike-wave (GSW) discharges, a characteristic electrographical feature of IGE, (3) locations of brain lesions associated with focal epilepsy, and (4) locations of scalp EEG electrodes.

The worldwide ENIGMA study identified significant atrophy in the bilateral precentral gyri and the thalamus[17] We tested whether these same brain regions were part of the identified IGE network by calculating the average network overlap (%) of the Desikan-Killiany[111] masks (i.e., the sum of the values in the IGE network within the mask divided by the number of voxels in each mask and multiplied with 100%). We compared this network overlap value to a null distribution of 10,000 randomly selected masks from the same atlas to assess significance. To test whether the identified IGE network may help explain the brain areas activated during generalized-onset epileptiform discharges, we searched the literature to identify simultaneous (inter)ictal EEG-fMRI studies in generalized and focal epilepsies (Supplementary Table 6). These studies were not included in the initial coordinate network mapping analysis because they lack a comparison to healthy controls, as is typical of EEG-fMRI analysis. We identified 12 studies from 10 publications that used the timings of epileptiform discharge onsets on scalp EEG in a general linear model to localize brain areas showing blood-oxygen-level-dependent activation in fMRI during these discharges (Supplementary Table 7). The results of these studies were separated into coordinates derived from six studies including patients with IGE and six other studies including patients with focal epilepsies (four temporal lobe epilepsy [TLE] studies and two extra-TLE studies). Again, we computed binary spheres (6 mm diameter) at each reported coordinate and generated study-level regions-of-interest (ROIs) to represent the brain areas of simultaneous EEG-fMRI activation. The average network overlap value within the IGE network was computed and compared between the ROIs from generalized epilepsy studies and focal epilepsy studies using a two-sample $t$-test. We hypothesized that EEG-fMRI activation associated with GSW discharges in patients with IGE would show higher overlap than EEG-fMRI activation associated with interictal epileptiform discharges in focal epilepsies. We compared alignment between the IGE network and a previously identified brain network derived from brain lesions associated with focal epilepsy[30] using a spatial correlation. Lesion locations were overlapped with the IGE network, and the average network overlap was compared between lesions associated with epilepsy ($n = 347$) and lesions not associated with epilepsy ($n = 1126$) using a two-sample $t$-test. To investigate the alignment of the IGE network with locations of scalp EEG, we created binary spheres (25 mm diameter) centered at the MNI coordinates of scalp electrodes placed according to the international 10–20 EEG system[41] and calculated overlap with the IGE network.

## Alignment with the somato-cognitive action network

Recently, ref. 42, revisited the topology and function of the motor cortex using resting-state fMRI. They showed that the motor cortex contains effector-specific regions (foot, hand, mouth) that are interrupted by inter-effector regions[42] involved in coordinating whole-body movement. Inter-effector areas were highly connected to a (sub)cortical network termed the somato-cognitive action network (SCAN), crucial for the integration of action planning with whole-body control.

As generalized tonic-clonic seizures are typically associated with involuntary movements of the whole body and limbs, we tested the alignment between our IGE network and the (inter)effector regions coordinating movement as defined by ref. 42. Spheres representing coordinates of the "inter-effector" and "effector" regions (Supplementary Tables 8, 9) were created at the published MNI coordinates[42] and the average overlap with the IGE network was compared using a $t$-test. We repeated this analysis using ROI-to-ROI connectivity analyses between the study level IGE coordinates and (inter)effector regions in each subject of our human connectome and compared the functional connectivity between these regions. Next, spheres located at the peak nodes of the SCAN were used as a seed in the same functional connectome used in the above coordinate network mapping analysis to create a whole-brain map of the SCAN. This map was then used to compute a spatial correlation between the IGE network and the SCAN. Finally, we compared ROI-to-ROI connectivity of the study-level IGE coordinates to the peak nodes of the SCAN and other previously defined canonical brain networks[112] and tested for a statistical difference across networks using a one-way ANOVA.

## Relevance to deep brain stimulation

To investigate the therapeutic relevance of the identified IGE network, we collected clinical outcome and imaging data from patients with IGE treated with CM DBS for the treatment of drug-resistant generalized seizures[43,44,113,114]. DBS electrodes were localized in MNI space using Lead-DBS software (https://www.lead-dbs.org)[115] and projected on top of the IGE network. Reduction in seizure frequency at the last seen follow-up moment was calculated as a percentage of change from before DBS surgery (Supplementary Table 10). To illustrate the potential clinical relevance of these network results to inform image-guided DBS therapy, we identified a patient with IGE from our center at the Brigham and Women's Hospital that received CM DBS. The DBS electrodes were localized and plotted in relation to the IGE network peak in the thalamus, previously published optimal DBS sites ("sweetspot") for IGE[44] and LGS[45], and discriminative fiber tracts[91] associated with improved control of generalized seizures after DBS.

## A convergent generalized epilepsy network

A CM DBS network was computed by (1) generating the volume of activated tissue (VAT, i.e., stimulation site) of each patient using patient-specific stimulation parameters[116], (2) identifying the brain networks connected to these DBS sites (VATs) using the same normative functional connectome, and (3) calculating the weighted average network across patients, weighted by each patient's clinical outcome (% reduction in seizure frequency) consistent with previous work[116]. The resulting CM DBS network was compared to the IGE network using a spatial correlation. Finally, a convergent IGE network was generated by $z$-scoring the IGE and CM DBS networks and averaging across these two networks.

## Statistical analysis

Statistical analyses were performed in R version 4.2.3 and MATLAB version 2018a (MathWorks). Non-parametric permutation tests were used to calculate $p$ values. A two-sided $p$ value $<0.05$ was considered significant, and we corrected it for multiple testing. Data were collected and analyzed from September 2019 through November 2023.

## Reporting summary

Further information on research design is available in the Nature Portfolio Reporting Summary linked to this article.

## Data availability

All coordinates of gray matter atrophy and fMRI hyperactivity used in this study are available in the published studies listed in supplementary materials. A version of the GSP connectome along with pre-processing details is publicly available[105]. The IGE network is available on github (https://github.com/jigongjun/IGENetwork/blob/main/IGEnetwork.nii). Source data are provided with this paper.

## Code availability

Code to conduct preprocessing and connectivity analyses is available as part of the open-access Lead-DBS software package (https://lead-dbs.org)[115] and WhiteMatterSF software (https://github.com/jigongjun/Neuroimaging-and-Neuromodulation; https://doi.org/10.5281/zenodo.14252802). Voxel-wise imaging results were projected onto a publicly available ultra-high resolution ex vivo brain aligned to MNI space[117].

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

## Acknowledgements

This study was funded by the National Natural Science Foundation of China, Grant Numbers: 82371507 (G.-J.J.), 32071054 (Y.T.), 82090034 (K.W.), U23A20424 (K.W.), and 31970979 (K.W.); the collaborative innovation project between universities and Hefei Comprehensive National Science Center (GXXT-2022-028, G.-J.J.); Hefei comprehensive national science center Hefei brain project (K.W.); Anhui Province Clinical Medical Research Transformation Special Project (202204295107020006 and 202204295107020028 to K.W.); Anhui Province Outstanding Youth Fund (2024AH020004 to G.-J.J.); Anhui province key research and development project (202104j07020033 to K.W.); the Science Fund for Distinguished Young Scholars of Anhui Province (1808085J23); F.L.W.V.J.S was supported by the American Epilepsy Society (846534) and National Institutes of Health (R01NS127892). L.J.D. and J.S.A. were supported by NHMRC project grants #628725 and #1108881, as were the imaging acquisition and prior analysis of DBS data related to LGS. J.D.R. and A.E.L.W. were supported by an NIH/NINDS grant (UH3NS109557A1). A.L.C. was supported by the NIMH (K23MH120510) and the Simons Foundation Autism Research Initiative. S.L. was funded by the Canadian Institutes of Health Research (CIHR) Banting Postdoctoral Fellowship. A.H. was supported by the German Research Foundation (Deutsche Forschungsgemeinschaft, 424778381 – TRR 295), Deutsches Zentrum für Luft- und Raumfahrt (DynaSti grant within the EU Joint Program Neurodegenerative Disease Research, JPND), the National Institutes of Health (R01 13478451, 1R01NS127892-01, 2R01 MH113929, and UM1NS132358) as well as the New Venture Fund (FFOR Seed Grant). M.D.F. was supported by grants from the National Institute of Mental Health, the National Institute on Aging, the Ellison-Baszucki Family Foundation, Kaye Family Research Endowment, and Manley family outside the submitted work. Dr. Fisher receives support from NIH

UG3NS114438, the James and Carrie Anderson Fund for Epilepsy, The Steve Chen Epilepsy Research Fund, and the Pilliod Research Fund. J.J. receives support from the Finnish Medical Foundation, Sigrid Juselius Foundation, Instrumentarium Research Foundation, and Turku University Hospital (VTR funds). Genomics Superstruct Project (GSP) data were provided (in part) by the Brain Genomics Superstruct Project of Harvard University and Massachusetts General Hospital (principal investigators: Randy Buckner, Joshua Roffman, and Jordan Smoller), with support from the Center for Brain Science Neuroinformatics Research Group, the Athinoula A. Martinos Center for Biomedical Imaging, and the Center for Human Genetic Research. Twenty individual investigators at Harvard University and Massachusetts General Hospital generously contributed data to the overall project. The preprocessed GSP connectome can be found at: https://doi.org/10.7910/DVN/ILXIKS%27, https://doi.org/10.7910/DVN/ILXIKS. Some of the data used in the preparation of this article were obtained from the ABCD study (https://abcdstudy.org), held in the National Institute of Mental Health Data Archive. This is a multisite, longitudinal study designed to recruit more than 10,000 children aged 9–10 years and follow them over 10 years into early adulthood. The ABCD study is supported by the NIH and additional fed- eral partners under award numbers U01DA041048, U01DA050989, U01DA051016, U01DA041022, U01DA051018, U01DA051037, U01DA050987, U01DA041174, U01DA041106, U01DA041117, U01DA041028, U01DA041134, U01DA050988, U01DA051039, U01DA041156, U01DA041025, U01DA041120, U01DA051038, U01DA041148, U01DA041093, U01DA041089, U24DA041123, and U24DA041147. A full list of supporters is available at https://abcdstudy.org/federal-partners.html. A listing of participating sites and a complete listing of the study investigators can be found at https://abcdstudy.org/scientists/ workgroups/. ABCD consortium investigators designed and implemented the study and/or provided data but did not necessarily participate in the analysis or writing of this report. This article reflects the views of the authors and may not reflect the opinions or views of the NIH or ABCD consortium investigators. The ABCD data repository grows and changes over time. The ABCD data used in this report are listed at: https://nda.nih.gov/study.html?id=1054, https://doi.org/10.15154/1520630. A.H. was supported by the German Research Foundation (Deutsche Forschungsgemeinschaft, 424778381 – TRR 295), Deutsches Zentrum für Luft- und Raumfahrt (DynaSti grant within the EU Joint Programme Neurodegenerative Disease Research, JPND), the National Institutes of Health (R01 13478451, 1R01NS127892-01, and 2R01 MH113929) as well as the New Venture Fund (FFOR Seed Grant). E.H.M. reports grant funding from Vigil Neuroscience, Inc. and the National Institutes of Health (U01-DK140734).

## Author contributions

G.-J.J. and F.L.W.V.J.S. conceptualized and designed the study, analyzed data and wrote the manuscript. Y.W. and M.M. analyzed data and edited the manuscript. All the authors have contributed to recruiting the patients and collected clinical samples and neuroimaging data. All authors (G.-J.J., M.D.F., M.M., Y.W., J.S., P.H., X.C., Y.J., C.Z., Y.T., Z.Z., H.A., J.N., J.J., C.T.D., S.G., G.G., M.T., L.J.D., J.S.A., R.S., I.S., A.V., J.Y., F.I., R.E.G., S.P., N.M.G., A.C., E.H.M., N.U.F.D., J.T., A.E.L.W., M.M.J.C., A.L.C., S.L., C.N., A.H., R.A.S., E.J.B., R.S.F., J.D.R., K.W., and F.L.W.V.J.S.) reviewed the final manuscript. M.D.F. and K.W. supervised the study, reviewed and edited the manuscript.

## Competing interests

M.D.F. serves as inventor on an active patent (US010137307B2) by Beth Israel Deaconess Medical Center that covers use of brain connectivity imaging to guide brain stimulation unrelated to the current work issued with no royalties and an active patent (US11666219B2) by Beth Israel Deaconess Medical Center that covers lesion network mapping (a precursor of coordinate network mapping used in this work) issued with no royalties. L.J.D. reports lecture fees for Boston Scientific. R.F. owns stock or options in Avails Medical, Cerebral Therapeutics, Eysz, Irody, Smart Monitor, and Zeto. A.H. reports lecture fees for Boston Scientific and is a consultant for FxNeuromodulation and Abbott. J.J. reports grants from the Research Council of Finland, Finnish Medical Foundation, Sigrid Juselius Foundation, Signe and Ane Gyllenberg Foundation, Finnish Foundation for Alcohol Studies, and Turku University Hospital; lecturer honoraria from Lundbeck and Novartis; travel support from Insightec, Abbvie, and Abbott, and consulting fees from Summaryx and Adamant Health; stocks of Neurologic Finland and Suomen Neurolaboratorio. E.H.M. serves as a consultant for Boston Scientific Corp, Olea Medical Inc., Varian Medical Systems Inc., and Cortechs.ai. Lecturer for Varian Medical Systems Inc. and Siemens Healthineers. Advisory Board member for Boston Scientific Corp and Varian Medical Systems Inc. The remaining authors declare no competing interests.

## Additional information

[1]Department of Neurology, The First Affiliated Hospital of Anhui Medical University, Anhui Medical University, Hefei, Anhui Province 230032, China. [2]Department of Psychology and Sleep Medicine, The Second Affiliated Hospital of Anhui Medical University, Anhui Medical University, Hefei 230032, China. [3]Anhui Province Key Laboratory of Cognition and Neuropsychiatric Disorders, Hefei 230032, China. [4]The School of Mental Health and Psychological Sciences, Anhui Medical University, Hefei 230032, China. [5]Anhui Institute of Translational Medicine, Hefei 230032, China. [6]Center for Brain Circuit Therapeutics, Department of Neurology, Neurosurgery, Psychiatry, and Radiology, Brigham and Women's Hospital, Harvard Medical School, Boston, USA. [7]Department of

Diagnostic Radiology, Jinling Hospital, the First School of Clinical Medicine, Southern Medical University, Nanjing 210002, China. [8]Queen Square Institute of Cognitive Neuroscience, University College London, London, UK. [9]Neurocenter, Department of Clinical Neurophysiology, Turku University Hospital, Turku, Finland. [10]Turku Brain and Mind Center, Clinical Neurosciences, University of Turku, Turku, Finland. [11]Department of Neurourgery, Hospital Universitario La Princesa, Universidad Autónoma de Madrid, Madrid, Spain. [12]Movement Disorders and Neurostimulation, Department of Neurology, Focus Program Translational Neuroscience (FTN), University Medical Center of the Johannes Gutenberg University Mainz, Rhine Main Neuroscience Network (rmn2), Mainz, Germany. [13]Department of Neurology, Hospital Universitario La Princesa, Universidad Autónoma de Madrid, Madrid, Spain. [14]Department of Medicine (Austin Health), The University of Melbourne, Victoria, Australia. [15]Department of Neurology, Boston Children's Hospital, Harvard Medical School, Boston, USA. [16]Department of Neurosurgery, King's College Hospital NHS Foundation Trust, London, UK. [17]Department of Basic and Clinical Neuroscience, King's College London, Institute of Psychiatry, Psychology and Neuroscience, London, UK. [18]Department of Clinical Neurophysiology, King's College Hospital NHS Foundation Trust, London, UK. [19]Department of Clinical Neurophysiology, Alder Hey Children's Hospital Trust, Liverpool, UK. [20]Department of Neurological Surgery, The Ohio State University College of Medicine, Columbus, OH, USA. [21]Department of Neurosurgery, Emory University, 1365 Clifton Road NE, Suite B6200, Atlanta, GA 30322, USA. [22]Departments of Neurosurgery, Emory University School of Medicine, Atlanta, Georgia, USA. [23]Department of Neurosurgery, Robert Wood Johnson Medical School, Rutgers University, New Brunswick, NJ 08901, USA. [24]Department of Neurology, Mayo Clinic, Rochester, MN, USA. [25]Department of Neurosurgery, São Paulo, Brazil. [26]Department of Radiology, Mayo Clinic, Jacksonville, FL, USA. [27]Mallinckrodt Institute of Radiology, Washington University School of Medicine, St Louis, MO, USA. [28]Department of Neurology, Washington University School of Medicine, St Louis, MO, USA. [29]Department of Biomedical Engineering, Washington University in St. Louis, St Louis, MO, USA. [30]MGH Neurosurgery & Center for Neurotechnology and Neurorecovery (CNTR) at MGH Neurology Massachusetts General Hospital, Harvard Medical School, Boston, MA 02114, USA. [31]Department of Neurology and Neurological Sciences and Neurosurgery by courtesy, Stanford University School of Medicine, Palo Alto, California, USA. [32]Institute of Artificial Intelligence, Hefei Comprehensive National Science Center, Hefei 230088, China. [33]These authors contributed equally: Gong-Jun Ji, Michael D. Fox, Kai Wang, Frederic L. W. V. J. Schaper. ✉e-mail: wangkai1964@126.com; fredericschaper@icloud.com

