## [Transparent Peer Review file · Nature Communications]

A generalized epilepsy network derived from brain abnormalities and deep brain stimulation

Corresponding Author: Professor Gong-Jun Ji

Version 0:

Reviewer comments:

Reviewer #1

(Remarks to the Author)

Ji, Fox and colleagues describe the detection of a generalised epilepsy network using coordinate network mapping. The authors assessed the connectivity of altered brain areas in voxel-based morphometry (VBM) or fMRI studies and validated the results using areas activated by generalised discharges in simultaneous EEG-fMRI. The manuscript is well written, interesting, timely, and novel. The results were robust across a number of analyses and provide meaningful data not only for neuroscientists but also for clinicians working with epilepsy.

Comments:

- The manuscript does not seem to provide a direct link with the brain network for lesional epilepsy, recently demonstrated by the authors. I suggest to put more emphasis on the similarities and differences between these two networks and also, if possible, provide a direct (statistical) comparison of both networks. Please comment on this in the results and discussion sections.
- The VBM studies included in the coordinate network mapping had small sample sizes. How do the results of these studies compare to larger studies, e.g. from the ENIGMA cohort (Whelan et al. 2018)? How do the structural abnormalities identified in the ENIGMA cohort map onto the brain network described in the current study?
- VBM may be less likely to identify cortical abnormalities, compared to surface based methods (i.e. cortical thickness measurements). Could this have an impact on the findings? Could the coordinate network mapping be extended to studies that analysed surface-based data?

Reviewer #2

(Remarks to the Author)

This is a well written manuscript on epileptic networks in IGE. They suggest that their findings derive from brain abnormalities and deep brain stimulation.

Their main functional, imaging, and anatomical data presentation and analysis refers to IGE. On the other hand, the major burden of the manuscript is that their DBS analysis was based on DEEs (mainly LGS, not IGE). The Melbourne series is LGS only, and the Spanish series, which is quoted as IGE, is actually a mix of patients (likely at least half of it is not IGE). There are other misquotations in the manuscript (like using a case report on VIM-DBS for focal epilepsy in the discussion of generalized epilepsy). It is inadequate to evaluate IGE data in the light of DEE (and vice-versa); they very likely have different pathophysiological mechanisms to explain their very different clinical presentation. They might consider keeping the IGE data and analysis only while revising the manuscript. Evaluating a single patient that was rendered seizure-free after DBS is likely non-relevant; this is not the most expected outcome from CM-DBS anyway.

Other issues:

In figure 5D right, the contacts appear to be outside CM (likely VIM) in the responder patient which disagrees with the authors' discussion.

In figure 6, the sweet spot appear to project into the VIM, and the electrode is clearly outside it, which disagrees with the authors' discussion.

Version 1:

Reviewer comments:

Reviewer #1

(Remarks to the Author)

Thank you for responding to the Referees' comments in full.

Response to reviewers:

A generalized epilepsy network derived from brain
abnormalities and deep brain stimulation

Ji & Fox *et al.*

Reviewer's Comments

Reviewer #1:

Ji, Fox and colleagues describe the detection of a generalised epilepsy network using coordinate network mapping. The authors assessed the connectivity of altered brain areas in voxel-based morphometry (VBM) or fMRI studies and validated the results using areas activated by generalised discharges in simultaneous EEG-fMRI. The manuscript is well written, interesting, timely, and novel. The results were robust across a number of analyses and provide meaningful data not only for neuroscientists but also for clinicians working with epilepsy.

Author response

We thank the reviewer for their kind comments and great suggestions which have significantly improved the quality of our manuscript.

Comments:

1. The manuscript does not seem to provide a direct link with the brain network for lesional epilepsy, recently demonstrated by the authors. I suggest to put more emphasis on the similarities and differences between these two networks and also, if possible, provide a direct (statistical) comparison of both networks. Please comment on this in the results and discussion sections.

Author response

We performed the requested analysis and found a negative spatial relationship ($R = -0.470$) between the network from Schaper et al. 2023¹ and the IGE network identified in this study, suggesting they are different networks and (partly) inverted. We calculated the overlap of the stroke lesions causing focal epilepsy from Schaper et al. 2023 with the IGE network and found they fall within the anticorrelations of the IGE network, while stroke lesions not causing epilepsy fall within the positive connections of the IGE network ($P < 0.001$). These findings suggest brain atrophy in IGE may be a consequence rather than a cause of epilepsy, consistent with diminished brain atrophy after temporal lobe resections in temporal lobe epilepsy² and previous brain atrophy findings in multiple psychiatric diseases³. To highlight this new finding, we have included a new Figure panel (Figure 3C) copied below. We have revised the methods, results, and discussion accordingly.

<<<<<

Methods

"We compared alignment between the IGE network and a previously identified brain network derived from brain lesions associated with focal epilepsy³⁰ using a spatial correlation. Lesion locations of strokes were overlapped with the IGE network and the average network overlap was compared between lesions associated with epilepsy ($n =$

76) and lesions not associated with epilepsy (n = 490) using a two-sample t test."

>>>>

Figure 3C. Lesion locations (white-black outlines) associated with epilepsy overlap with negative functional connections in the IGE network, while lesions not associated with epilepsy (controls) overlap with positive functional connections (C).

<<<<<

Results

"We investigated alignment between the IGE network and a previously identified brain network derived from brain lesions associated with focal epilepsy³⁰ and found they were inversely related ($r = -0.470$). Lesions associated with epilepsy overlapped with negative functional connections in the IGE network, while lesions not associated with epilepsy overlapped with positive functional connections ($t = 11.05$, $P < 0.001$)."

>>>>

<<<<<

Discussion

"Focal features in IGE predominantly localize to the temporal lobe⁶⁷, consistent with the topography of negative functional connections of the IGE network and a brain network derived from lesions associated with focal epilepsy. This IGE network may therefore provide a potential mechanism to explain generalized epilepsy with clinical and

electrographic focal features^{71,72}.”

“Second, it remains unclear whether brain abnormalities such as atrophy in IGE are a cause, consequence, or compensatory mechanism of epilepsy. Several interpretations could be made. They could reflect changes in a network damaged by seizures and thus be a direct consequence of seizures. They could also reflect abnormal brain development that is causing seizures. Other hypotheses suggest atrophy is part of a degenerative process that facilitates epilepsy, an effect of antiseizure drugs, or some combination of these factors⁹⁷. A different potential interpretation may be that atrophy reflects changes in a network attempting to inhibit or compensate for seizures. The finding that brain lesions associated with focal epilepsy are anticorrelated to the brain regions atrophied in IGE suggests they involve different brain networks and that atrophy may be a consequence or compensatory mechanism in epilepsy rather than a cause, consistent with previous network mapping findings^{30,98,37}. Future prospective longitudinal studies are needed to elucidate the causal cascade of these structural network changes in relationship to clinical outcome and the spatiotemporal dynamics of seizures within this network.”

>>>>

2. The VBM studies included in the coordinate network mapping had small sample sizes. How do the results of these studies compare to larger studies, e.g. from the ENIGMA cohort (Whelan et al. 2018)? How do the structural abnormalities identified in the ENIGMA cohort map onto the brain network described in the current study?

Author response

We performed the requested analysis and found that the structural abnormalities in IGE identified in the worldwide ENIGMA study map onto the IGE network identified in the current study. Whelan et al. 2018 investigated structural changes in IGE patients compared to healthy controls in a worldwide multicenter ENIGMA study⁴. They found cortical atrophy in three regions of interest (ROIs) of the Desikan-Killiany atlas: the bilateral precentral gyri and the thalamus. All three regions identified in the ENIGMA study are part of our IGE network. Overlap of these three regions with the IGE network identified in this study is significantly higher compared to a null distribution of randomly selecting 3 ROIs of the Desikan-Killiany atlas and recomputing network overlap 10,000 times ($P = 0.015$). To highlight this new finding, we have included a new Figure panel (Figure 3A) copied below and methods, results, and discussion sections were revised accordingly.

<<<<

Methods

“The worldwide ENIGMA study identified significant atrophy in the bilateral precentral gyri and the thalamus.¹⁷ We tested whether these same brain regions were part of the identified IGE network by calculating the average network overlap (%) of the Desikan-

Killiany¹¹¹ masks (i.e. the sum of the values in the IGE network within the mask divided by the number of voxels in each mask and multiplied with 100%). We compared this network overlap value to a null distribution of 10,000 randomly selected masks from the same atlas to assess significance.”

>>>>

Figure 3A. Brain regions atrophied in IGE (white outlines) as identified in the worldwide ENIGMA study were part of the IGE network and overlapped more with this network compared to a null distribution of randomly selected brain regions

<<<<

Results

“The worldwide ENIGMA study identified significant atrophy in the bilateral precentral gyri and the right thalamus 17. These same brain regions were part of the identified IGE network and overlapped more with the network compared to a null distribution of regions randomly selected from the same atlas ($t = 0.53$, $P = 0.015$, Figure 3A).”

>>>>

<<<<

Discussion

“The IGE network comprised cortical and subcortical areas including positive functional connectivity to SMA, sensorimotor cortex, anterior cingulate, superior temporal gyrus, piriform cortex, thalamus, basal ganglia, and cerebellum; and negative functional connectivity (“anticorrelation”^{46,47}) to the frontal poles, medial frontal lobe, angular gyrus, precuneus, middle and inferior temporal gyri. These regions have previously been

implicated in the generation of spike and wave discharges in animal models of generalized epilepsy^{48,49}, align with structural abnormalities in IGE previously identified in the worldwide ENIGMA study¹⁷ and results from other surface based neuroimaging studies in patients with IGE^{6,7,10,17,18,50,51}. "

>>>>

3. VBM may be less likely to identify cortical abnormalities, compared to surface based methods (i.e. cortical thickness measurements). Could this have an impact on the findings?

The reviewer is correct that both surface- and volume-based metrics can be used to identify structural changes in patients versus controls. In a previous study, Tetreault et al. 2020⁵ found that the brain network identified by coordinate network mapping using volume based data from VBM studies was almost identical to the brain network identified using surface based methods. This suggests volume and surface based data identify the same brain network. Our finding that surface-based atrophy locations identified in the worldwide ENIGMA study map onto our IGE network supports this notion (see Reviewer 1 comment 2, and newly included Figure 2C).

We would like to highlight that in the current study, we specifically chose to use only volume-based data rather than surface-based data for three important reasons: 1) IGE is associated with atrophy in the subcortex which can be overlooked by surface based methods and gross volume measurements of atlas masks, 2) we prioritized being consistent with the methods of previously published coordinate network mapping papers using volume based VBM data and volume based fMRI connectomes, 3) the coordinates for the volume-based studies are commonly shared in a table within the published paper allowing open access to these coordinates, while for surface based methods, we would need the original patient images which were unavailable to us.

4. Could the coordinate network mapping be extended to studies that analysed surface-based data?

Author response

Yes, the method can be extended to surface based data and more specifically vertex wise surface data. This has previously been investigated by Tetreault et al. 2020⁵ and is part of ongoing work from our group (La Rivière et al. 2023, accepted at JAMA Neurology⁶). It remains to be determined whether the IGE network identified in this study using VBM data would be replicated using vertex-wise surface data from patients with IGE, which is a topic of future work. We have included a comment in the discussion on the need for method development in this area.

<<<<<

Discussion

“Future studies using single-subject level data in combination with modern volume- and surface-based analysis methods could more precisely identify structural abnormalities in patients. Such efforts may help identify specific networks involved in different IGE sub-syndromes, seizure types, or even patient-specific seizure networks^{91,92}.”

>>>>

Reviewer #2:

This is a well written manuscript on epileptic networks in IGE. They suggest that their findings derive from brain abnormalities and deep brain stimulation. Their main functional, imaging, and anatomical data presentation and analysis refers to IGE. On the other hand, the major burden of the manuscript is that their DBS analysis was based on DEEs (mainly LGS, not IGE).

Author response

We thank the reviewer for providing helpful feedback to improve the quality of our manuscript. We agree that the analyses on structural brain abnormalities mainly refers to IGE, while the DBS data is mainly based on LGS. As recommended by the reviewer, we have removed the LGS data from the manuscript and collected new data of CM DBS in 21 patients with IGE from 6 different centers worldwide.

1. The Melbourne series is LGS only, and the Spanish series, which is quoted as IGE, is actually a mix of patients (likely at least half of it is not IGE). It is inadequate to evaluate IGE data in the light of DEE (and vice-versa); they very likely have different pathophysiological mechanisms to explain their very different clinical presentation. They might consider keeping the IGE data and analysis only while revising the manuscript.

Author response

We agreed with our Spanish collaborators to have a board-certified epileptologist re-review the clinical and EEG data of all 10 CM-DBS patients that were included in our submitted manuscript. Consistent with the reviewer's comment, we found that the CM-DBS patients are a mix of LGS and other DEEs, and none could be diagnosed with IGE at time of DBS in contrast to our previous communication and initial submission. We greatly thank the reviewer for catching this issue and we have therefore removed this Spanish DEE dataset in conjunction with the Australian LGS dataset from our manuscript as recommended by the reviewer.

To test whether CM-DBS stimulation sites from IGE patients intersect our IGE network, we collected new data from patients with IGE (n=21) that were not included in our initial submission. To the best of our knowledge, we received most published (n=14) and several unpublished (n=7) cases of CM DBS in patients with IGE from 6 different centers worldwide. We found that 1) the CM DBS sites from patients with IGE highly align with our IGE network peak in the thalamus derived from brain abnormalities in IGE, 2) the whole brain connectivity of these DBS sites weighted by clinical outcome recapitulates the IGE network derived from brain abnormalities in IGE, 3) the IGE network peak in the central thalamus aligns more with a published CM DBS sweetspot derived from patients with IGE compared to a published CM DBS sweetspot derived from patients with LGS, consistent with the comment raised by the reviewer that the optimal DBS sites and networks for IGE and LGS are likely different.⁷ Our findings converge on a common brain network for brain abnormalities in IGE and effective CM DBS sites in IGE, may help explain why the CM is effective at reducing generalized seizures in patients with IGE, and support the CM as a DBS target for patients with IGE.

<<<<<

Methods

"Relevance to deep brain stimulation

To investigate the therapeutic relevance of the identified IGE network, we collected clinical outcome and imaging data from patients with IGE treated with CM-DBS for the treatment of drug-resistant generalized seizures^{41,42,113,114}. DBS electrodes were localized in MNI space using Lead-DBS software (<https://www.lead-dbs.org>)¹¹⁵ and projected on top of the IGE network. Reduction in seizure frequency at the last seen follow-up moment was calculated as a percentage of change from before DBS surgery (Supplementary Table 10). To illustrate the potential clinical relevance of these network results to inform image-guided DBS therapy, we identified a patient with IGE from our center at the Brigham and Women's Hospital that received CM-DBS. The DBS electrodes were localized and plotted in relation to the IGE network peak in the thalamus, previously published optimal DBS sites ("sweetspot") for IGE⁴² and LGS⁴³, and discriminative fiber tracts⁸² associated with improved control of generalized seizures after DBS.

A convergent generalized epilepsy network

A CM DBS network was computed by 1) generating the volume of activated tissue (VAT, i.e., stimulation site) of each patient using patient-specific stimulation parameters¹¹⁶, 2) identifying the brain networks connected to these DBS sites (VATs) using the same normative functional connectome, and 3) calculating the weighted average network across patients, weighted by each patient's clinical outcome (% reduction in seizure frequency) consistent with previous work¹¹⁶. The resulting CM DBS network was compared to the IGE network using a spatial correlation. Finally, a convergent IGE network was generated by z-scoring the IGE and CM DBS networks and averaging across these two networks."

>>>>

Figure 5. Relevance for deep brain stimulation. DBS electrode locations implanted at the CM to treat drug resistant generalized seizures in patients with IGE were localized with Lead-DBS software (A). The IGE network peaked in the CM (B), which was the most functionally connected nucleus in the thalamus (C). Seizures reduced a median 90% after CM DBS on patients with IGE (D) and DBS electrodes intersected with the peak of the IGE network in the CM (E).

<<<<<

Results

Relevance to deep brain stimulation

"The IGE network involved peak functional connectivity to the centromedian nucleus (CM) of the thalamus, a region that has been used as a DBS target to treat generalized seizures⁴¹ (Figure 5A-C). We analyzed the DBS electrode locations and clinical outcome (% reduction in seizure frequency) of 21 patients with IGE that were treated with CM-DBS for drug resistant generalized seizures. Seizure frequency reduced a median of 90% (interquartile range: 66.5-96.5%), 66.7% on average (standard error: 14.3) and 19 of 21 patients (90%) were responders with >50% reduction in seizure frequency considered clinically significant. DBS electrode locations aligned with the topography of the IGE network in the thalamus (Figure 5E). Notably, the IGE network peaked in the CM (Figure 6A) converging in the same location (MNI coordinate: $x = -9.05$, $y = -21.07$, $z = -0.07$) as a recently identified optimal DBS site ("sweetspot") for IGE⁴² (Figure 6B), and 4 mm closer

compared to a DBS sweetspot for patients with LGS⁴³ (Figure 6C). To illustrate the clinical potential of these findings to inform image-guided DBS for generalized epilepsy, we localized the DBS electrodes of an independent patient with IGE treated in our center at the Brigham and Women's Hospital and visualized the IGE network peak, the previously reported DBS sweetspots in patients with IGE or LGS, and discriminative fibers of the reticular system (Figure 6D-E) associated with improved generalized seizure control after CM DBS."

Figure 6. Illustration of the potential clinical translation for image-guided DBS. The DBS electrode locations of an independent patient with IGE treated with CM-DBS was plotted in relation to the IGE network peak in the thalamus (A, red), the previously reported DBS sweetspots in IGE (B, green) and LGS (B, purple), and discriminative fiber tracts associated with improved generalized seizure control after CM DBS (C, pink). The IGE network peak converged on a similar location to these optimal DBS sites, yet 4 mm closer to the sweetspot derived from IGE versus LGS patients.

A convergent generalized epilepsy network

The IGE network (Figure 7A) derived from coordinates was spatially similar (spatial $r = 0.67$) to a CM DBS network (Figure 7B) derived from the networks of the individual patient's DBS sites weighted by clinical outcome (% reduction in seizure frequency). A convergent IGE network was identified by averaging the IGE network and CM DBS network (Figure 7C)."

>>>>

Figure 7. A convergent generalized epilepsy network. The IGE network (A) was spatially similar to the CM DBS network (B) and a convergent IGE network was identified (C).

<<<<

Discussion

"Our findings may also have therapeutic implications for brain stimulation treatment in generalized epilepsy. Specifically, they suggest that the CM nucleus in the thalamus is a key node in an IGE network consistent with recordings of epileptiform discharges in the CM region during generalized seizures⁷⁹⁻⁸¹ and early involvement of the CM during GSW discharges in EEG-fMRI. These network results may help explain the positive results of CM-DBS and RNS in patients with IGE as reported by case- and open-label studies. Brain networks connected to neuroimaging abnormalities in IGE thus converge on a DBS target used to treat generalized seizures^{43,82}. Serendipitous, the IGE network peaked in the same location in the thalamus as an optimal DBS site ("sweetpot") for patients with IGE⁴² and a CM DBS network derived from the stimulation sites of patients with IGE

recapitulated the IGE network. Overall, these findings suggest potential clinical utility of the IGE network to identify brain stimulation targets and could be used to guide DBS, responsive neurostimulation (RNS) or non-invasive brain stimulation therapies such as transcranial magnetic stimulation, multifocal transcranial electric stimulation, and focused ultrasound⁸³ "

>>>>

2. There are other misquotations in the manuscript (like using a case report on VIM-DBS for focal epilepsy in the discussion of generalized epilepsy).

Author response

The reviewer is correct, the case report discusses a focal epilepsy patient. We have removed the quotation.

3. Evaluating a single patient that was rendered seizure-free after DBS is likely non-relevant; this is not the most expected outcome from CM-DBS anyway.

Author response

The reviewer is correct, seizure freedom is not the most expected outcome after CM-DBS. We have removed the notion that the example patient is seizure free throughout the manuscript. The intended message of Figure 6 is to illustrate the locations of the IGE network peak in reference to the different optimal DBS sites (sweetspots) that were previously reported and visualize a case example of how our results could inform image-guided DBS for generalized epilepsy in the future, akin to recent developments in movement disorders⁸. The manuscript and figure legend were revised to clarify this without emphasizing seizure freedom in this single patient as suggested by the reviewer.

<<<<<

Figure 6. Illustration of the potential clinical translation for image-guided DBS. The DBS electrode locations of an independent patient with IGE treated with CM-DBS was plotted in relation to the IGE network peak in the thalamus (A, red), the previously reported DBS sweetspots in IGE (B, green) and LGS (B, purple), and discriminative fiber tracts associated with improved generalized seizure control after CM DBS (C, pink). The IGE network peak converged on a similar location to these optimal DBS sites, yet 4 mm closer to the sweetspot derived from IGE versus LGS patients.”

>>>>>

Other issues:

4. In figure 5D right, the contacts appear to be outside CM (likely VIM) in the responder patient which disagrees with the authors' discussion.

Author response

We have removed the notion that this patient is a responder and only use the DBS electrode of this case to illustrate how our results could inform image-guided DBS in generalized epilepsy.

5. In figure 6, the sweet spot appear to project into the VIM, and the electrode is clearly outside it, which disagrees with the authors' discussion.

Author response

The reviewer is correct, the CM DBS sweetspot identified in patients with LGS is in the VLPv/VIM atlas mask. During this revision process, a new CM DBS sweetspot identified in patients with IGE was published: Park et al. (2024) *Epilepsia*⁹. This sweetspot for IGE is within the CM atlas mask, and closer to our IGE network peak (Euclidean distance of 0.869 mm) compared to the DBS sweetspot for LGS (Euclidean distance of 5.065 mm), and also closer to the DBS electrode location of the case example illustrated in Figure 6. We have revised the methods, results, and discussion accordingly and revised Figure 6 to include this newly published IGE DBS sweetspot.

Figure 6. Illustration of the potential clinical translation for image-guided DBS. The DBS electrode locations of an independent patient with IGE treated with CM-DBS was plotted in relation to the IGE network peak in the thalamus (A, red), the previously reported DBS sweetspots in IGE (B, green) and LGS

(B, purple), and discriminative fiber tracts associated with improved generalized seizure control after CM DBS (C, pink). The IGE network peak converged on a similar location to these optimal DBS sites, yet 4 mm closer to the sweetspot derived from IGE versus LGS patients.

References

1. Schaper FLWVJ, Nordberg J, Cohen AL, et al. Mapping Lesion-Related Epilepsy to a Human Brain Network. *JAMA Neurol.* 2023;80(9):891-902. doi:10.1001/jamaneurol.2023.1988
2. Galovic M, de Tisi J, McEvoy AW, et al. Resective surgery prevents progressive cortical thinning in temporal lobe epilepsy. *Brain J Neurol.* 2020;143(11):3262-3272. doi:10.1093/brain/awaa284
3. Taylor JJ, Lin C, Talmasov D, et al. A transdiagnostic network for psychiatric illness derived from atrophy and lesions. *Nat Hum Behav.* 2023;7(3):420-429. doi:10.1038/s41562-022-01501-9
4. Whelan CD, Altmann A, Botía JA, et al. Structural brain abnormalities in the common epilepsies assessed in a worldwide ENIGMA study. *Brain J Neurol.* 2018;141(2):391-408.
5. Tetreault AM, Phan T, Orlando D, et al. Network localization of clinical, cognitive, and neuropsychiatric symptoms in Alzheimer's disease. *Brain J Neurol.* 2020;72:1048.
6. Larivière S, Schaper FLWVJ, Royer J, et al. Cortical alterations and responsive neurostimulation map to hippocampal brain networks in temporal lobe epilepsy. *JAMA Neurol.* 2024;in press. doi:10.1001/jamaneurol.2024.2952
7. Warren AEL, Tobochnik S, Chua MMJ, Singh H, Stamm MA, Rolston JD. Neurostimulation for Generalized Epilepsy: Should Therapy be Syndrome-specific? *Neurosurg Clin N Am.* 2024;35(1):27-48. doi:10.1016/j.nec.2023.08.001
8. Rajamani N, Friedrich H, Butenko K, et al. Deep brain stimulation of symptom-specific networks in Parkinson's disease. *Nat Commun.* 2024;15(1):4662. doi:10.1038/s41467-024-48731-1
9. Park S, Permezel F, Agashe S, et al. Centromedian thalamic deep brain stimulation for idiopathic generalized epilepsy: Connectivity and target optimization. *Epilepsia.* Published online September 14, 2024. doi:10.1111/epi.18122